# Resolving Dirac electrons with broadband high-resolution NMR

Wassilios Papawassiliou [1], Aleksander Jaworski[1], Andrew J. Pell [1✉], Jae Hyuck Jang[2], Yeonho Kim[2], Sang-Chul Lee [2], Hae Jin Kim [2✉], Yasser Alwahedi [3,4], Saeed Alhassan[3], Ahmed Subrati [3,5], Michael Fardis[6], Marina Karagianni [6], Nikolaos Panopoulos[6], Janez Dolinšek[7] & Georgios Papavassiliou[6✉]

Detecting the metallic Dirac electronic states on the surface of Topological Insulators (TIs) is critical for the study of important surface quantum properties (SQPs), such as Majorana zero modes, where simultaneous probing of the bulk and edge electron states is required. However, there is a particular shortage of experimental methods, showing at atomic resolution how Dirac electrons extend and interact with the bulk interior of nanoscaled TI systems. Herein, by applying advanced broadband solid-state $^{125}$Te nuclear magnetic resonance (NMR) methods on $Bi_2Te_3$ nanoplatelets, we succeeded in uncovering the hitherto invisible NMR signals with magnetic shielding that is influenced by the Dirac electrons, and we subsequently showed how the Dirac electrons spread inside the nanoplatelets. In this way, the spin and orbital magnetic susceptibilities induced by the bulk and edge electron states were simultaneously measured at atomic scale resolution, providing a pertinent experimental approach in the study of SQPs.

[1] Department of Materials and Environmental Chemistry, Arrhenius Laboratory, Stockholm University, Svante Arrhenius vag 16C, SE-106 91 Stockholm, Sweden. [2] Electron Microscopy Research Center, Korea Basic Science Institute, 169-148 Gwahak-ro, Yuseong-gu, Daejeon 34133, Republic of Korea. [3] Department of Chemical Engineering, Khalifa University, PO Box 2533Abu Dhabi, UAE. [4] Center for Catalysis and Separation, Khalifa University of Science and Technology, P.O.Box 127788Abu Dhabi, UAE. [5] NanoBioMedical Centre, Adam Mickiewicz University, Wszechnicy Piastowskiej 3, 61-614 Poznań, Poland. [6] Institute of Nanoscience and Nanotechnology, National Center for Scientific Research "Demokritos", 153 10, Aghia Paraskevi, Attiki, Greece. [7] J. Stefan Institute and University of Ljubljana, Faculty of Mathematics and Physics, Jamova 39, SI-1000 Ljubljana, Slovenia. ✉email: andrew.pell@mmk.su.se; hansol@re.kbsi.kr; g.papavassiliou@inn.demokritos.gr

n the presence of spin–orbit coupling (SOC), the spin **s** and the orbital angular momentum **l** of the electron lose their time invariance; it is the total angular momentum of the electron **j** = **s** + **l** that preserves it. This fundamental property of electrons is the playground of a number of fascinating phenomena, such as the Dirac edge states in topological insulators (TIs)[1], the quantum spin Hall effect[2], and the formation of Majorana fermions[3–5]. In case of three-dimensional TIs, SOC forces the surface electrons to form helical spin structures, wrapping an odd number of massless Dirac cones, with the simplest and most-studied systems being the $Bi_2Se_3$ and $Bi_2Te_3$ tetradymides. Angle resolved photoemission spectroscopy (ARPES) experiments in combination with theoretical studies have shown that these systems acquire a single Dirac Cone and large band gap[6,7], thus providing an ideal platform for studying SQPs. However, despite the simplicity of their topological edge states, important quasiparticle excitations of Dirac electrons, such as the Majorana zero mode[3–5], the excitonic superfluid condensate[8], or the propagation of chiral spin waves on topological surfaces[9] remain to a great extent experimentally unexplored. It is furthermore noticed that the detection of Majorana excitations requires the simultaneous probing of the response from both the bulk and edge states, which is difficult to achieve with current experimental techniques[10].

Since many SQPs depend on the way that the spin of the Dirac electrons couples with their orbital motion and how this interaction propagates through the crystal, an experimental probe sensitive to both the spin and orbital motion of the Dirac electrons with atomic scale resolution is crucial in the efforts to further understand the physics of topological materials. NMR appears to fulfill these requirements as the nuclear magnetic shielding, and consequently the NMR frequency shift, depends on the spin and orbital magnetic susceptibility at the position of each resonating nucleus. Specifically, the total NMR Knight shift can be expressed as $\delta_{tot} = \delta_{FC} + \delta_{dip} + \delta_{orb}$, where the first two terms are the Fermi-contact and spin-dipolar terms originating from the electron spin polarization at the Fermi level, and the third is the orbital term generated by the orbital currents of the Dirac electrons[11,12]. According to recent theoretical calculations on TI's the orbital term $\delta_{orb}$ from the Dirac electrons dominates over $\delta_{FC}$ and $\delta_{dip}$, and induces large negative shifts, and very short spin-lattice relaxation times $T_1$[12]. However, until now experimental $^{125}Te$ NMR studies on $Bi_2Te_3$ nanoparticles[13] and microcrystalline (bulk) powders[14] have given contradicting results.

In this article, combining advanced broadband solid-state NMR methods with $C_s$-corrected scanning transmission electron microscopy (STEM) and DFT calculations on pristine $Bi_2Te_3$ nanoplatelets, we succeeded for the first time to resolve the NMR signal assigned to the Dirac electron states, and monitored through the NMR parameters the way that the spin and orbital magnetic susceptibilities induced by the Dirac electrons vary across the nanoplatelets.

## Results

### Crystal and band structure analysis of the $Bi_2Te_3$ nanoplatelets.

$Bi_2Te_3$ nanoplatelets were synthesized in the liquid phase using a solvothermal method, as described in the "Methods" section. Figure 1a and Supplementary Figs. 1–3 illustrate the excellent quality of the nanoplatelets, which exhibit perfect hexagonal shapes with sharp edges, average diameter of 600 nm, and mean thickness of 10 nm. The structural characteristics were examined at atomic scale by means of high angle annular dark/bright field HAADF/ABF imaging, as presented in Fig. 2a and Supplementary Fig. 3. Since the intensity of the HAADF images is proportional

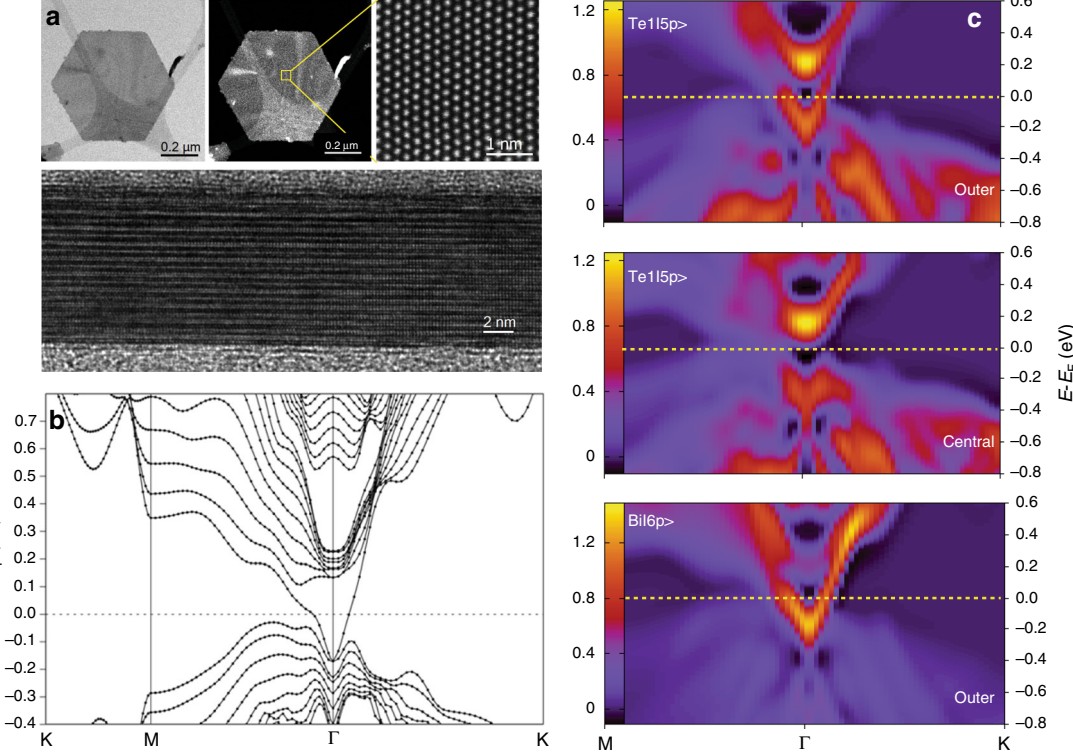

**Fig. 1 Band structure analysis and Dirac states of stoichiometric $Bi_2Te_3$ nanoplatelets. a** High-resolution TEM and HAADF image of $Bi_2Te_3$ nanoplatelets (top view and cross-section). **b** Band structure of a 9-quintuplets-thick (~10 nm) $Bi_2Te_3$ slab. **c** The projected $k$-resolved DOS of the Te(1) |5p⟩ and Bi |6p⟩ orbital states at the outer (edge) quintuplets, and the Te(1) |5p⟩ orbital states at the central quintuplet. Dirac states are observed only at the edge quintuplets (more details in Supplementary Fig. 6).

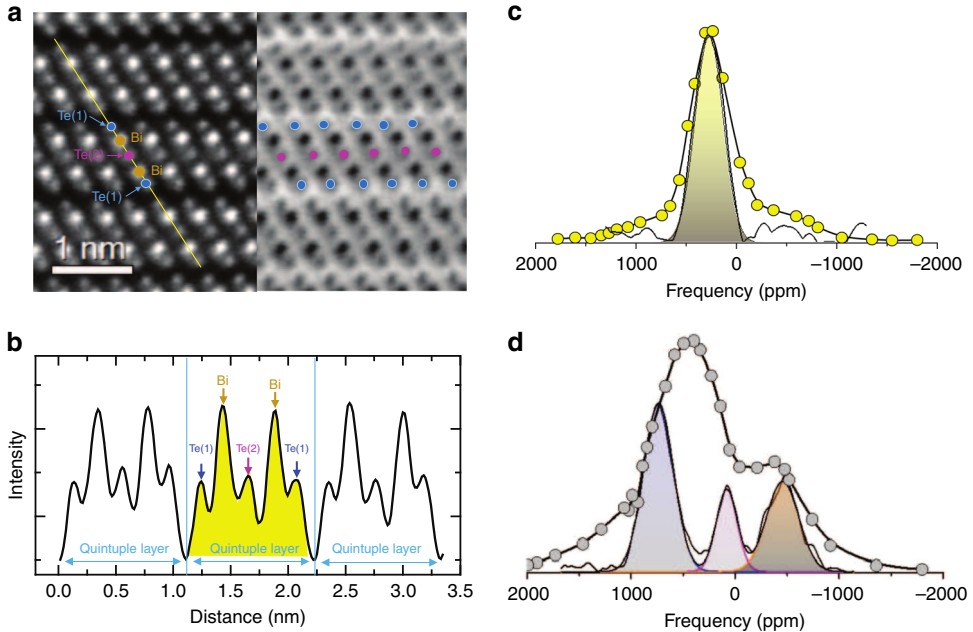

**Fig. 2 Atomic scale TEM analysis and 1D $^{125}$Te NMR. a** Cross-sectional HAADF-ABF images of a $Bi_2Te_3$ nanoplatelet. Brown, blue, and magenta dots indicate the Bi, Te(1) and Te(2) columns in the quintuplets. **b** The intensity profile from the cross section (yellow line) in the HAADF image. **c** $^{125}$Te static frequency sweep NMR (yellow circles) and the isotropic projection of the 2D $^{125}$Te aMAT NMR at 14 kHz MAS (solid line) of bulk $Bi_2Te_3$. **d** $^{125}$Te static frequency sweep NMR (gray circles) and the isotropic projection of the 2D $^{125}$Te aMAT at 30 kHz MAS of a $Bi_2Te_3$ nanoplatelets sample.

to the atomic number $Z^2$, atomic columns in the HAADF/ABF image could be identified, and were labeled according to the Bi, Te(1), and Te(2) atomic sites by brown, blue, and magenta dots, respectively[15]. The Bi and Te atoms are organized in well-defined quintuple atomic layers comprising of five covalently bonded atomic sheets of alternating Bi and Te atoms, i.e. Te(1)–Bi–Te (2)–Bi–Te(1), that are bound to each other by van der Waals interactions, as seen in the intensity profiles in Fig. 2b and Supplementary Fig. 3.

In order to examine the way in which the Dirac states propagate through the nanoplatelets and how this feature is encoded into the NMR Knight shift, DFT calculations were carried out on a $Bi_2Te_3$ slab comprising 9 quintuplets, corresponding to the mean thickness of the $Bi_2Te_3$ nanoplatelets. Details on the DFT calculations are given in the Supplementary Information (SI). Figure 1b shows the relevant band structure, in agreement with previous calculations[6]. In particular, the band structure shows an energy band gap of $\approx160$ meV and a single surface Dirac cone enclosing the $k = \Gamma$ symmetry point. The Fermi energy was calculated at $E_F = 6.5942$ eV, i.e. at slightly higher energy than the maximum of the valence band. Figure 1c, and Supplementary Figs. 4–6 show that the Dirac states are defined mainly by the Te(1) $|5p\rangle$ and Bi $|6p\rangle$ states of the terminating quintuplets. Furthermore, the density of the Dirac states is observed to be substantially reduced in the central region of the nanoplatelets. It is therefore expected that the NMR signals from the surface of the nanoplatelets will be shifted with respect to the signals from the bulk (interior), because the Dirac electrons are predicted to induce large negative orbital Knight shifts[12]. Despite this expectation, until now efforts to detect the Dirac states through $^{125}$Te NMR have not been successful, either with static or with magic-angle-spinning (MAS) one-dimensional (1D) NMR methods. The main reason is the large Knight shift anisotropy, which gives rise to broad unresolved NMR signals that are difficult to excite.

The shortcomings of 1D NMR for acquiring well-resolved spectra is clearly seen in Fig. 2c, d, which compare the static frequency-sweep $^{125}$Te NMR spectra of two samples (c) microcrystalline (bulk) $Bi_2Te_3$ where Dirac electron states are almost absent and (d) $Bi_2Te_3$

nanoplatelets where Dirac electron states dominate in the electron band structure. Both spectra exhibit different overall profiles, but with similar features; each is characterized by a central peak at a shift of 250 ppm (microcrystals) and 480 ppm (nanoplatelets), respectively, with a tail at higher shift, and a shoulder at negative shift. In case of the microcrystalline material these features have been explained as originating from the presence of two overlapping signals shifted relative to each other[14]: one strong narrow resonance at 250 ppm from Te(1), and a broader asymmetric resonance at a shift of −400 ppm from Te(2). Implementation of advanced 1D solid-state MAS NMR methods, such as the double adiabatic echo (DAE) experiment did not improve the resolution, as seen in the Supplementary Fig. 8c.

**Resolving the bulk and Dirac edge states with aMAT NMR.** In order to resolve these individual $^{125}$Te NMR signals, and to identify those from the topological edge states, the 2D adiabatic magic-angle turning (aMAT) NMR experiment was implemented (Fig. 3a), which in the indirect dimension provides the isotropic NMR shifts free from spectral broadening due to any kind of anisotropy[16,17]. Details on the implemented NMR techniques are provided in the "Methods" section. The effectiveness of the method is in evidence in Fig. 2c, d, which overlay the isotropic $^{125}$Te NMR projections of the 2D aMAT of both the microcrystalline and nanoplatelet samples onto the 1D frequency-sweep spectra. In the case of the microcrystalline sample a single broad resonance is observed at 250 ppm, which comprises the two overlapping Te(1) and Te(2) NMR signals. No isotropic signal component is observed at negative shifts, as is also clearly shown in Supplementary Fig. 7. In case of the nanoplatelets the isotropic "bulk" NMR signal resolves into two distinct components at shifts of 765 and 93 ppm with a ratio of integrals of 2:1, corresponding to Te(1) and Te(2) respectively; The difference in the isotropic NMR shift between the microcrystalline and nanoplatelets samples reflects differences in the electron/hole doping of the two samples[18], and in general in the distribution of the conduction electrons across the nanoplatelets. Most importantly, a third distinct signal component is observed in

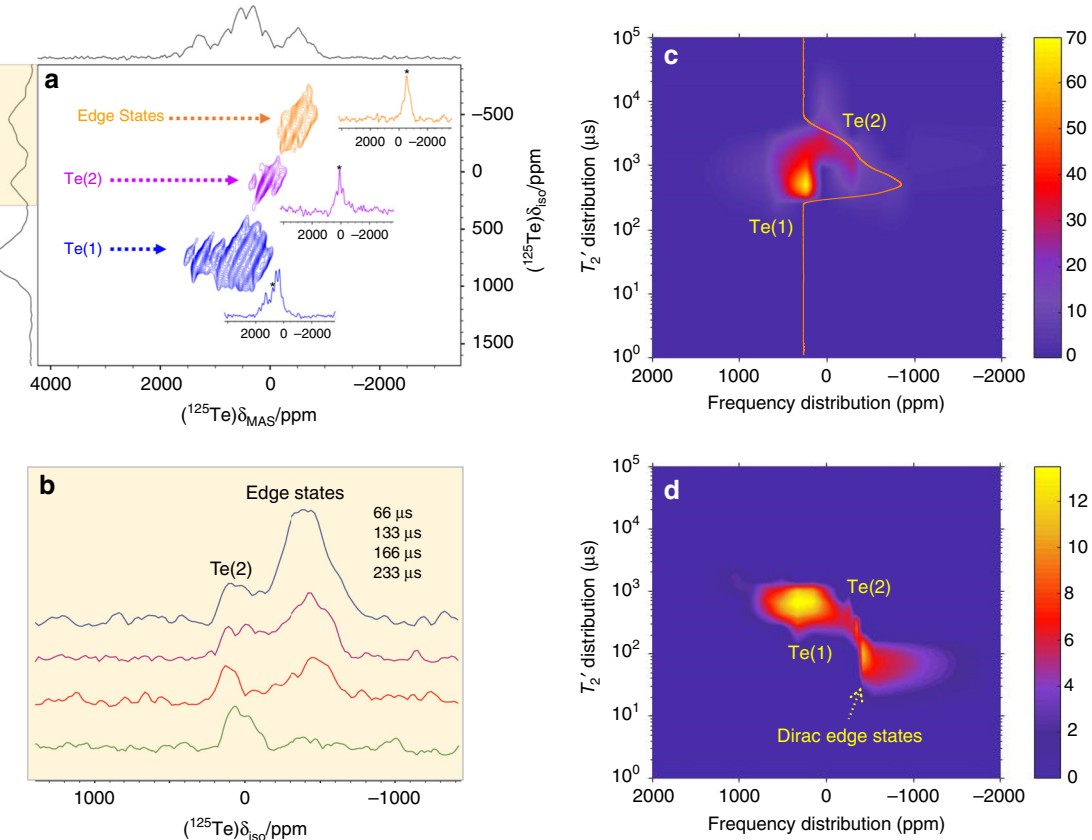

**Fig. 3 $^{125}$Te aMAT NMR and $T_2'$ dephasing analysis of the Dirac edge states. a** 2D $^{125}$Te aMAT NMR spectrum of $Bi_2Te_3$ nanoplatelets. Blue and magenta color contours indicate signals from the bulk interior of the nanoplatelets, while the orange color contours show signals from the surface Te sites, shielded by the orbital motion of the Dirac electrons. **b** The expanded isotropic projections of $^{125}$Te MAT NMR spectrum acquired at four different evolution times. **c** The $^{125}$Te NMR $T_2'$ distribution as a function of the resonance frequency of microcrystalline (bulk) $Bi_2Te_3$. The orange color cross section shows the $T_2'$ distribution at frequency 250 ppm. **d** The $^{125}$Te NMR $T_2'$ distribution with respect to the resonance frequency of the $Bi_2Te_3$ nanoplatelets.

the aMAT NMR signal of the nanoplatelets at a markedly different shift of −452 ppm, which is assigned to the topological edge states. This signal appears reproducibly in different nanoplatelet samples, including those exhibiting some oxidation of the surface, as observed in Supplementary Fig. 8, which is an important finding for many applications. Furthermore, the possibility that this signal is produced by trivial edge effects, such as (i) sudden changes in the surface composition or structure (e.g. stacking faults) and (ii) gradual structural variations can be ruled out. In the first case, with the aid of TEM and high-resolution cross-sectional HAADF imaging (Supplementary Figs. 2 and 3), we see clean surfaces with no evidence of defects, thus confirming that the nanoplatelets are pristine. In the second case, we would expect substantial inhomogeneous broadening of bulk NMR signals, rather than the additional resonance at −452 ppm.

We note that, because of the long adiabatic pulses (33.33 μs)[17], aMAT signals with very short spin–spin relaxation times $T_2$ may be partially wiped out. The way that nuclear spin coherences dephase across the spectrum is shown in Fig. 3c, d, which display the $T_2'$ distribution vs. shift for both the microcrystalline and nanoplatelet samples, acquired by 1D inverse-Laplace-transform Carr–Purcell–Meiboom–Gill (CPMG) spin-echo pulse trains.

In addition to the inherent $T_2$ relaxation, $T_2'$ dephasing includes coherent signal decay due to the extended nuclear dipolar coupling network across the particles[19]. Experimental details on the CPMG inversion are presented in the SI. In the case of the microcrystalline sample the two signals corresponding to Te(1) and Te(2) are resolved, with Te(2) showing a highly anisotropic

frequency distribution. In case of the nanoplatelets, the third strong signal component at −452 ppm, exhibits significantly shorter $T_2'$ times, indicating enhanced relaxation, which is attributed to the Dirac electrons.

To test whether such short $T_2'$ influences the signal intensity in the aMAT spectrum, the isotropic shift projections of standard MAT experiments were obtained at different shift evolution times as shown in Fig. 3b[16,17,20]. The large signal intensity at −452 ppm at short evolution times indicates that a large proportion of the atomic layers in the nanoplatelets have a surface-like electronic structure, and its rapid dephasing at longer evolution times relative to the other signals from the nanoplatelet interior confirms the difference in dephasing times shown in Fig. 3d.

On comparing the $T_2'$ distribution maps for the microcrystalline and nanoplatelet samples in Fig. 3c, d, we note two striking features: firstly, the $T_2'$ coherence lifetimes of the signals due to bulk Te have the same value of ~1 ms; and secondly the $T_2'$ of the signal at −452 ppm for the nanoplatelet sample is an order of magnitude shorter, at ~0.1 ms. Whilst the interpretation of these $T_2'$ data is complicated by the fact that $T_2'$ encapsulates the effects of both spin–spin relaxation, and coherent dephasing effects, these observations are consistent with enhanced relaxation of the surface Te due to the Dirac electrons according the following mechanisms. The enhanced $T_2'$ relaxation may be assigned to slow orbital magnetic field fluctuations[21], produced by the surface orbital currents of the Dirac electrons, or otherwise to the Ruderman–Kittel–Kasuya–Yosida (RKKY) interaction between the nuclear spins, also mediated by the Dirac electrons[22].

The variation of the spin-lattice relaxation times $T_1$ with the Knight shift is somewhat different, as shown in Supplementary Fig. 11. Here, we see that the measured $T_1$s are uniformly an order of magnitude shorter for the nanoplatelets compared to the microcrystalline sample (200 vs. 20 ms), but see no variation between the bulk Te compared to the surface Te signals, within experimental error, as we do for $T_2'$. This is an interesting observation, which indicates that in these quasi-2D topological systems the nuclear spin relaxation mechanisms are more intricate than considered[12]. Nevertheless, the observation of a shorter $T_1$ for the nanoplatelets is evidence that the Dirac electrons prevail on the spin-lattice relaxation mechanism, the effect of which extends into the bulk. This uniformity for $T_1$ across the whole nanoplatelet may be explained by the quasi-2D topology of the $Bi_2Te_3$ nanoplatelets[12], which confines orbital currents in parallel to the external surfaces. In Fermi liquids with similar orbital current topology, Lee and Nagaosa[23] have shown that $T_1$ is independent of the position of the resonating nuclei, which resembles our observation.

**DFT calculations.** To strengthen our interpretation of the $^{125}Te$ aMAT spectra, and in particular to confirm the assignment of the resonance at −452 ppm, DFT calculations of the NMR shifts were carried out, using the full-potential linearized augmented plane-wave method, as implemented in the Wien2k DFT software package. Figure 4a, c show the calculated orbital, Fermi-contact (spin), and spin-dipolar terms of the Knight shift in the presence of SOC, of bulk $Bi_2Te_3$ and a 5-quintuplets slab (with thickness $d \sim 5$ nm), representing the microcrystalline $Bi_2Te_3$ and the

nanoplatelets, respectively. The latter was selected because it allows tractable DFT calculations with satisfactory $k$-grid, while Dirac electron states remain gapless[24]. The total isotropic Knight shift is, as previously discussed, equal to $\delta_{tot} = \delta_{FC} + \delta_{dip} + \delta_{orb}$. Conventionally, the orbital term is referenced according to $\delta_{orb} = \sigma_{ref} − \sigma_{orb}$[25], with the reference shielding set to $\sigma_{ref} = 2370$ ppm, from Supplementary Fig. 9, whilst the Fermi-contact and the spin dipolar terms, $\delta_{FC}$ and $\delta_{dip}$, are set as minus the relevant calculated shielding, i.e., $−\sigma_{FC}$, respectively $−\sigma_{dip}$. In case of the microcrystalline material $\delta_{tot}$ was found to be +720 ppm for Te (1) and −660 ppm for Te(2), as shown in Fig. 4a, c. These values are shifted in comparison to the experimental ones, mainly due to the strong temperature dependence of the NMR frequency[14], as shown in the inset of Fig. 4a. (Data in the inset of Fig. 4a are not isotropic NMR Knight shifts, but merely the frequency position of Te(1) and Te(2) NMR peaks, acquired from 1D $^{125}Te$ NMR spectra.) However, we note that this variation with temperature is similar for the shifts of both Te(1) and Te(2) and, in particular, that the ordering of the two peaks does not change. It is stressed that in the presence of SOC, calculations of NMR shifts require substantial computational efforts, and are dependent on intrinsic errors of the GGA-DFT level of theory, number of $k$-points, and smearing of the electron occupancy around the Fermi level[11]. Nevertheless, the calculated NMR shifts, and the predicted ordering of the peaks in the spectrum, are sufficient to facilitate the interpretation of the experimental NMR signals.

In the case of the 5-quintuplet-thick slab the isotropic total Knight shift of Te(1) and Te(2) coalesce to a single resonance, acquiring strong negative shift, $\delta_{tot} \sim −2670$ ppm. This is

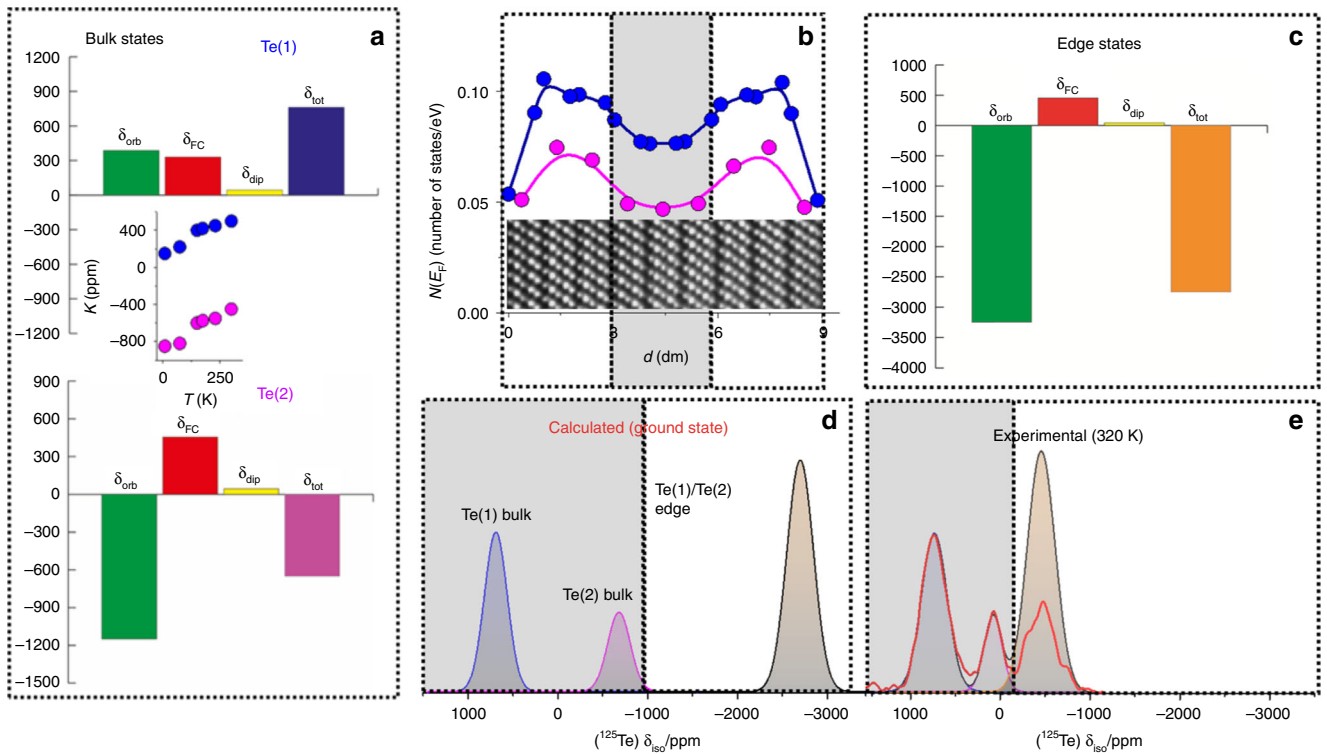

**Fig. 4 DFT analysis of the $^{125}Te$ NMR Knight shifts in $Bi_2Te_3$ nanoplatelets. a** Calculated $^{125}Te$ orbital (green), Fermi contact (red), and dipolar terms (light yellow) of the Knight shift of bulk $Bi_2Te_3$ (SOC is set on). The inset shows the temperature dependence of the Knight shift according to experimental data in ref. [14]. **b** The Te(1) and Te(2) DOS at the Fermi level $N(E_F)$, across a 9-quintuplet (~9 nm) $Bi_2Te_3$ slab. The gray-shaded area marks a central region in the slab with "bulk-like" character, i.e. significant drop of the DOS. **c** Calculated $^{125}Te$ Knight shift of a 5-quintuplet $Bi_2Te_3$ slab (SOC is set on). **d** Simulation of the $^{125}Te$ NMR signal of a 9-quintuplet slab, by combining calculated bulk and edge state Knight shifts. The bulk/edge intensity ratio was set according to the number of Te sites in in the central (gray-shaded) and edge areas in panel **b**. Calculated NMR lines were convoluted with a Gaussian function. **e** The experimental isotropic $^{125}Te$ NMR spectrum of the $Bi_2Te_3$ nanoplatelets. The NMR signal from the edge states at −452 ppm was corrected to account for differential $T_2'$ signal dephasing.

explainable by the fact that in very thin nanoplatelets ($d \leq 5$ nm), Dirac electrons spread almost uniformly across the slab (Supplementary Fig. 5), inducing a strong negative Knight shift $\delta_{orb}$ via their orbital currents. By increasing the slab thickness the system splits into an interior "bulk" region that occupies the center of the slab, which is sandwiched between two "topological" surface regions. This is shown in Fig. 4b, which displays the Te(1) and Te(2) $N(E_F)$ across a 9-quintuplets slab. At the center of the nanoplatelet the DOS drops significantly (gray shaded area) and a band gap begins to open, as shown in Fig. 1c and Supplementary Fig. 6. Combining the theoretical bulk and surface Knight shifts at the appropriate signal intensity ratio, as explained in the caption of Fig. 4, the simulated ground state $^{125}$Te NMR spectrum of a 9-quintuplet-thick slab was depicted (Fig. 4d), and compared with the experimental isotropic $^{125}$Te NMR spectrum of the nanoplatelets (Fig. 4e). In the latter case, the shown signal intensities have been corrected by considering the $T_2'$ dependence of the signal on the interpulse time intervals of the aMAT experiment. In analogy to the microcrystalline system, the calculated NMR Knight shifts are more negatively shifted in comparison to the experimental ones, due to the strong, temperature dependent, negative shift; however the similar NMR patterns of the experimental and calculated spectra are unambiguous evidence that the NMR signal at $-452$ ppm originates from the Dirac electron states.

## Discussions

Bearing in mind the strong negative NMR frequency shift by decreasing temperature[12,14], a nice correlation between the theoretical and experimental spectra is witnessed. This is strong evidence that the isotropic $^{125}$Te NMR signal component at $-452$ ppm belongs to the surface NMR signal shielded by the Dirac electrons. In this perspective, the intensity ratio of the surface-to-bulk NMR signals provides the mean volume of the nanoplatelets that is occupied by the Dirac electrons, whereas $T_1$ (Supplementary Fig. 11) and $T_2'$ relaxation measurements on the pertinent peaks highlight the interaction of the Dirac electrons with the bulk interior. While ultrathin $Bi_2Te_3$ nanoplatelets are shown to be quantum topological species, where the NMR signals are dominated by the Dirac electrons, it is expected that by increasing the thickness of the nanoplatelets the NMR signals of the bulk interior will prevail, while at intermediate thicknesses, the interaction of the Dirac electrons with the bulk interior can be monitored both through the change in the $T_1$ and $T_2'$ times of the distinct NMR peaks, and the measured Knight shifts. The presented experimental approach enables the simultaneous probing of the bulk and edge states, which is important in the study of SQPs.

## Methods

**Synthesis**. $Bi_2Te_3$ nanoplatelets were synthesized following a solvothermal approach. Specifically, 1 mmol $BiCl_3$ and 1.5 mmol $Na_2TeO_3$ were dispersed in 15 mmol of an alkaline solution (NaOH), and 1.16 M polyvinylpyrrolidone (PVP, $M_w = 40,000$ g/mol Da) were dissolved in 40 mL of ethylene glycol. The mixture was magnetically stirred until it turned highly translucent, then it was transferred and sealed into a Teflon-lined stainless-steel autoclave (capacity of 80 mL). The sealed autoclave was put into an oven at 180 °C for 36 h and cooled to room temperature. The resulting products were collected by repeated centrifugations, and subsequently washed with distilled water and ethanol, two times each, and finally vacuum dried overnight at 90 °C for further characterization.

**NMR**. The $^{125}$Te MAS experiments of the $Bi_2Te_3$ microcrystalline material were performed with a 4 mm HXY triple-resonance probe, at 14 kHz MAS on a Bruker 400 Avance-III spectrometer operating at a $^{125}$Te Larmor frequency of 126.23 MHz. Spectral acquisition was done with a double adiabatic spin-echo sequence (DAE) with a 2.5 μs 90° excitation pulse length, corresponding to an RF field of 100 kHz, followed by a pair of rotor-synchronized short high-power adiabatic pulses (SHAPs)[17] of 71.43 μs length and a 5 MHz frequency sweep. For the separation of the isotropic shift and chemical shift anisotropy, which is of significant magnitude in heavy spin-1/2 nuclei[26] and strong electron correlated systems, the adiabatic

magic-angle-turning (aMAT) pulse sequence was employed[16], which consists of a $\pi/2$ excitation pulse followed by six refocusing SHAP $\pi$-pulses. The same SHAPs as in the DAE were used. Separation of isotropic Knight Shifts was achieved in the isotropic dimension, whereas the MAS dimension corresponds to the conventional MAS spectrum. The diagonal ridges in the sidebands appear because of the inhomogeneous broadening of the sidebands, as they extend along the isotropic shift for each respective Te site. The $^{125}$Te MAS spectra of the $Bi_2Te_3$ nanoplatelets were acquired on the same spectrometer with a 2.5 mm HX probe, at 30 kHz MAS. For the acquisition of the DAE and the aMAT spectra, rotor synchronized SHAPs sweeping through 5 MHz in 33.33 μs were employed with RF field amplitude of 160 kHz. For the aMAT spectra, the same SHAPs were used with an evolution time of 66.66 μs, excluding the length of the SHAPs, which is equivalent to two rotor periods. Chemical shifts were referenced to $TeO_2$[27].

The frequency-sweep $^{125}$Te NMR spectra were acquired on a home-built NMR spectrometer under static conditions, operating at Larmor frequency of 126.23 MHz. For the spin-lattice relaxation time $T_1$ and the coherence lifetime $T_2'$ experiments a $\pi/2$-t-$\pi/2$ saturation recovery pulse sequence and a CPMG pulse sequence $\pi/2$-$\tau$-$\{\pi$-$2\tau$-$\pi$-...-$\pi$-$2\tau$-$\pi\}_{300}$ with a train of 300 $\pi$ pulses were implemented, respectively. The $T_1$ and $T_2'$ distribution analysis was performed by applying a non-negative Tikhonov Regularization Algorithm. Details on the relaxation analysis are provided in the SI.

**Electron microscopy (SEM/STEM)**. Scanning electron microscopy (SEM) images were recorded on a Hitachi S4800 microscope while the size distribution of the $Bi_2Te_3$ nanoplatelets were obtained by measuring 100 randomly selected $Bi_2Te_3$ platelets in the SEM image. STEM images and their energy-dispersive X-ray (EDX) elemental mappings were measured using a JEOL JEM-2100F. High-resolution TEM and STEM images were acquired using a Jeol ARM200 with probe $C_s$-corrector, operated at 200 kV. Cross-sectional TEM samples were prepared by FIB in KBSI (Quanta 3D FEG, FEI). Atomic-coordinates analysis from HAADF images was performed via intensity refinement method. The atomic coordinates analysis was performed with the following steps: (i) HAADF image normalization, (ii) Laplacian of Gaussian filtering, (iii) image erosion, (iv) atom position detection using circular pattern matching, (v) pattern matching errors correction, (vi) center position of atoms correction, (vii) computation of the average pixel intensity around the center of atoms, and finally (viii) two kind of markers based on intensity values has been depicted on the HAADF image, so that the coordinates of Bi and Te are clearly identified from each other.

**XRD**. The XRD spectra were recorded with an analytical PANalytical X'Pert PRO powder diffractometer. The sample was mounted on a zero-background holder and scanned by using Cu-Kα radiation ($\lambda = 1.5418$ Å) with the following experimental conditions: applied voltage of 40 kV, intensity of 30 mA, angular range ($2\theta$) 5–80° and 0.03 steps/s. Rietveld refinement of obtained powder XRD pattern was carried out using the FULLPROF program software. Refined parameters include: overall scale factor, background (BGP), lattice parameters, atomic positions and orientation.

**DFT calculations**. DFT calculations were carried out with the QUANTUM ESPRESSO package[28] on a $Bi_2Te_3$ slab comprised of 9 quintuplets, acquiring the mean thickness of the $Bi_2Te_3$ nanoplatelets. The slab surface was set according to the top-view HAADF image in Fig. 1a, along the (001) plane. Calculations were performed on the basis of the Perdew–Burke–Ernzerhof (PBE) type generalized gradient approximation. For the Brillouin zone integrations we used a $11 \times 11 \times 1$ Monkhorst–Pack $\mathbf{k}$-point mesh, and the kinetic energy cutoff was fixed to 800 eV. The lattice constants were acquired by the Rietveld refinement of the XRDs ($a = b = 4.395$ Å, $c = 29.830$ Å). Spin–orbit effects were treated self-consistently using fully relativistic projector augmented wave (PAW) pseudopotentials[29].

NMR Knight Shift calculations were performed by using the full-potential linearized augmented plane-wave method, as implemented in the Wien2k DFT software package[30]. The spin–orbit interaction was considered in a second variational method. Calculations were performed with and without SOC on two different atomic configurations: bulk $Bi_2Te_3$, and a five quintuplets $Bi_2Te_3$ slab (25 atoms) with thickness $\sim5$ nm. The $k$-mesh convergence was checked up to 100,000 points for the bulk materials and up to 5000 points for the slab. Other computational parameters like atomic sphere radii, as well as potentials and wave functions inside the atomic spheres are kept as set by Wien2k defaults. The plane wave basis set size was determined by seting $RK_{max} = 8$, and for presented results we have used the PBE generalized gradient approximation[31]. The orbital part of the Knight shift $\delta_{orb}$ was calculated by using the x_nmr script of the Wien2k software package, by activating switches to include SOC, the Dirac states metallicity, and Fermi–Dirac smearing between 2 and 8 mRy. The Fermi contact and dipolar terms were calculated in the presence of SOC, using a spin-polarized set-up in an external magnetic field of 100 T, as explained in refs. [11,30].

## Data availability

The authors declare that the data supporting the findings of this study are available within the article and its Supplementary Information file. Extra data are available from the corresponding authors upon reasonable request.

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

## Acknowledgements

W.P., A.J., and A.J.P. were supported by the Swedish Research Council (project no. 2016-03441). M.K. and G.P. acknowledge support by the project MIS 5002567, implemented under the "Action for the Strategic Development on the Research and Technological Sector", funded by the NSRF 2014–2020 and co-financed by the European Union and Greece. Part of the DFT work was performed using computational resources of the Research Computing Department at Khalifa University. Y.A. would like to acknowledge the support of Khalifa University of Science and Technology Award No. RC2-2018-024. Open access funding provided by Stockholm University.

## Author contributions

A.J.P., W.P., H.J.K., and G.P. conceived and designed the experiments, performed theoretical analysis, and did the majority of paper writing (with contribution from all coauthors). aMAT and MAT NMR experiments were performed by W.P., A.J., and A.J.P. Frequency sweep NMR and relaxation studies were performed by W.P., M.F., M.K., J.D., and N.P. DFT calculations were performed by W.P., Y.A., A.S., and G.P. Sample preparation and characterization was performed by Y.K., S.-C.L., and S.A. HRTEM experiments were performed by J.H.J., and H.J.K.

## Competing interests

The authors declare no competing interests.
