## [Peer Review File · Nature Communications]

Reviewers' comments:

Reviewer #1 (Remarks to the Author):

The article written by Papawassiliou and coauthors deals with a topological insulator Bi₂Te₃ and its surface "edge" states. Experimental methodology and material design utilised here are nicely sophisticated enough to detect spectrum that can be attributed to the edge-state. I highly admit an importance of their work that developed new technique to observe surface Dirac properties of nano-TIs, which would be highlight of the field.

In contrast to the experimental part, comprehension of the results seems to be lacking scientific rigor, though. To be correct, NMR-site assignments for Te bulk 1,2, and edge-states needs to be very accurate. However, due to strong SOC fashion of Te atom, DFT calculations end up with poor matches as shown in ref22 and SI. Even after performing the amazing factor-by-minus-one correction, the calculated spectrum works merely qualitatively. Details for the Wien2k calculations, necessary for reproduction, are also missing. Therefore, I recommend the authors to improve the discussion lines greatly. The authors tried to support their comprehension for NMR Knight shift, T₁, T₂ behavior in accord with ref 12. But again, significant errors are included. Ref 12 deals with 3D Dirac system, and then the details should be different. Shorter T₂ should not be referred as an evidence of Diracness because T₂ << T₁. And also T₁ for "edge-state" mentioned in SI is longer than those of Te_{1,2}. This contradicts to the statement given for T₂.

In conclusion, observation of Dirac edge-state may be correct if one look at the sample dependences as circumstantial evidences. However, the calculations did not help understanding of data but slightly misleading.

Reviewer #2 (Remarks to the Author):

Referee Report - NCOMMS-19-34029 - Papawassiliou et al. - Resolving Dirac electrons with broadband high-resolution NMR: New perspectives in exploring topological quantum properties

Summary:

The authors Papawassiliou et al. present magic angle turning NMR results on carefully characterized nano-platelets of Bi₂Te₃, which provide strong evidence for the observation of topological Dirac electrons. Specifically, the authors find a third resonance (different from the bulk Te(1) and Te(2) sites with a shift and spin-lattice relaxation that are consistent with topological surface electrons. DFT calculations support the authors interpretation of the data.

Recommendation:

This work is novel, of high quality, and provides strong evidence for the observation of surface topological states with NMR. Therefore, I recommend publication in nature communications following a few quick questions/comments.

Comments/Questions:

The authors have checked for surface reconstruction in both experiment and theory, and therefore seem to have ruled this possibility out, but do not seem to have explicitly stated that trivial surface effects could be ruled out. Is this the case? Are there other possible trivial effects that show up in other nano-crystalline materials that may explain the data?

Sample thickness dependence would be highly useful as a future direction. This would of course require significant further work on the synthesis side, and therefore is beyond the scope of this manuscript, but it would be best if the authors could use their existing data to comment/predict the

expected dependence of the NMR spectrum as the sample thickness is increased and the Dirac electrons no longer span the entire sample and the bulk states begin to dominate?

Minor:

What is the physical significance of the diagonal ridges (oriented from bottom left to top right) in the 2d spectra shown in fig 3a?

On page 6 the authors refer to the topological edge states as having a shift of -452 ppm and then two paragraphs later as having -500 ppm. It seems the second number is rounded? Perhaps it is better to use a consistent rounded or unrounded value so as not to confuse the reader.

Red and magenta are difficult to tell apart for Bi vs Te in the TEM figure etc. Perhaps reconsider using colors that are more different.

The authors sometimes label axes with "arbitrary units" and sometimes omit this label, e.g. 2b vs 2c,d. Perhaps consider a blanket statement or label all axes appropriately...

Can the authors give a better explanation of the DFT vs experiment referencing procedure in the SM? My understanding is that a few NMR standards are used to extract a slope and intercept, and use this intercept as the "standard" that characterizes the difference between DFT and experiment. Can the authors elaborate on this a bit more (just one or two sentences).

We are grateful to the reviewers for making a number of helpful comments, which we feel have led to the improvement of the manuscript. We have now revised the manuscript according to the points raised by both reviewers in their reports. Below, please find the original reviewer comments in blue colour and the authors' response in black. All changes outlined in the text below are highlighted in yellow colour in a highlighted version of the revised manuscript.

Reviewer #1 (Remarks to the Author):

The article written by Papawassiliou and coauthors deals with a topological insulator Bi_2Te_3 and its surface "edge" states. Experimental methodology and material design utilized here are nicely sophisticated enough to detect spectrum that can be attributed to the edge-state. I highly admit an importance of their work that developed new technique to observe surface Dirac properties of nano-TIs, which would be highlight of the field.

We would like to thank the reviewer for his very positive opinion regarding the novelty of the experimental work presented, and his valuable comments, which give us the opportunity to improve significantly the discussion on the main issues he/she raises.

In contrast to the experimental part, comprehension of the results seems to be lacking scientific rigor, though. To be correct, NMR-site assignments for Te bulk 1,2, and edge-states needs to be very accurate. However, due to strong SOC fashion of Te atom, DFT calculations end up with poor matches as shown in ref. 22 and SI. Even after performing the amazing factor-by-minus-one correction, the calculated spectrum works merely qualitatively.

We thank the reviewer for his comments. The symbols used to annotate the Knight shift in Figure 4 were not sufficiently explained, and the temperature dependence of the Knight shift, as well as the relation between calculated shielding values σ and Knight Shift required more discussion. We have revised the discussion (yellow highlighted text in the "DFT calculations" paragraph, in pages 9 and 10) and we now believe that the revised text meets the criteria of clarity set by the reviewer.

It is furthermore noticed that in the presence of SOC, calculated NMR shifts require laborious calculations, and are sensitive to the selected exchange-correlation functional, number of k -points, and smearing of the electron occupancy around the Fermi level. nevertheless, the calculated NMR shifts are sufficient to facilitate the interpretation of the experimental NMR signals.

Details for the Wien2k calculations, necessary for reproduction, are also missing. Therefore, I recommend the authors to improve the discussion lines greatly.

Text has been added in the DFT calculations paragraph of the Methods section (page 13 – highlighted yellow), providing details on the DFT calculations, which enable reproduction of the results.

The authors tried to support their comprehension for NMR Knight shift, T_1 , T_2 behavior in accord with ref 12. But again, significant errors are included. Ref 12 deals with 3D Dirac system, and then the details should be different. Shorter T_2 should not be referred as an evidence of Diracness because $T_2 \ll T_1$. And also T_1 for "edge-state" mentioned in SI is longer than those of $\text{Te}_{1,2}$.

This contradicts to the statement given for T_2 .

The Reviewer is correct to note that $T_1 \gg T_2$, and that there is no direct connection between the T_2' dephasing enhancement for the surface Te vs bulk Te due to the Dirac electrons with those data presented in Figure 3 with ref. 12, which concern the spin-lattice T_1 relaxation enhancement due to Dirac electron orbital fluctuations. In addition, in our case there is no T_1

relaxation enhancement at the surface of the nanoplatelets relative to the, as predicted in ref. 12 for 3D TI systems, but merely a T_2' dephasing enhancement.

In the revised version of our article, we have included two paragraphs (highlighted yellow in pages 8 and 9), which discuss these observations, and provide some possible explanations. We indicate that a T_2' dephasing enhancement is possible both by orbital magnetic field fluctuations, produced by the surface orbital currents of the Dirac electrons, and Ruderman-Kittel-Kasuya-Yosida (RKKY) indirect coupling of nuclear spins, mediated by Dirac electrons.

The uniformity of the T_1 relaxation times across the whole nanoplatelet may be explained by considering the quasi-2D topology of the Bi_2Te_3 nanoplatelets. In such cases, the orbital currents are confined in parallel to the external surfaces of the nanoplatelets. In the case of Fermi liquids with similar topology, Lee and Nagaosa (new ref. 23) have shown that T_1 is independent of the position of the resonating nuclei in the system, which resembles with our finding.

In conclusion, observation of Dirac edge-state may be correct if one look at the sample dependences as circumstantial evidences. However, the calculations did not help understanding of data but slightly misleading.

We anticipate that the revised discussion about the Knight shifts, DFT calculations, and relaxation mechanisms has addressed the concerns of the Reviewer, and that we have further highlighted the novelty of our experimental approach and shows that solid-state NMR can be used to follow up the evolution of the Dirac states by changing temperature, material thickness, surface structure, etc.

Reviewer #2 (Remarks to the Author):

Referee Report - NCOMMS-19-34029 - Papawassiliou et al. - Resolving Dirac electrons with broadband high-resolution NMR: New perspectives in exploring topological quantum properties

Summary:

The authors Papawassiliou et al. present magic angle turning NMR results on carefully characterized nano-platelets of Bi₂Te₃, which provide strong evidence for the observation of topological Dirac electrons. Specifically, the authors find a third resonance (different from the bulk Te(1) and Te(2) sites with a shift and spin-lattice relaxation that are consistent with topological surface electrons. DFT calculations support the authors interpretation of the data.

Recommendation:

This work is novel, of high quality, and provides strong evidence for the observation of surface topological states with NMR. Therefore, I recommend publication in nature communications following a few quick questions/comments.

We would like to thank the Reviewer for considering our article as novel and worth to be published in Nature Communications.

Comments/Questions:

The authors have checked for surface reconstruction in both experiment and theory, and therefore seem to have ruled this possibility out, but do not seem to have explicitly stated that trivial surface effects could be ruled out. Is this the case? Are there other possible trivial effects that show up in other nano-crystalline materials that may explain the data?

We have now added an explicit discussion on page 6 that rules out possible trivial explanations for our observations. In particular, we note that (i) sudden changes in the surface composition or structure (e.g. stacking faults) are ruled out by the TEM measurements and high-resolution cross-sectional HAADF imaging (Figure S2, Figure S3), where we see clean surfaces with no evidence of defects, thus confirming that the nanoplatelets are pristine. In addition we see that (ii) gradual structural variations can be ruled out, as here we would expect substantial inhomogeneous broadening of bulk NMR signals, rather than the additional resonance at -452 ppm.

Sample thickness dependence would be highly useful as a future direction. This would of course require significant further work on the synthesis side, and therefore is beyond the scope of this manuscript, but it would be best if the authors could use their existing data to comment/predict the expected dependence of the NMR spectrum as the sample thickness is increased and the Dirac electrons no longer span the entire sample and the bulk states begin to dominate?

The Reviewer is entirely correct to note this as a very interesting perspective for future studies. We have added some text to the conclusions illustrating the possible perspectives for future NMR on this topic (pages 10 and 11).

Minor:

What is the physical significance of the diagonal ridges (oriented from bottom left to top right) in the 2d spectra shown in fig 3a?

The diagonal ridges in the 2D aMAT NMR spectrum depict the inhomogeneous broadening of the sidebands (similar to the sidebands in conventional 1D MAS NMR spectra), which in 2D extends diagonally along the isotropic shift for each Te site. A short explanatory text has been added in pages 11-12 of the section NMR-methods.

On page 6 the authors refer to the topological edge states as having a shift of -452 ppm and then two paragraphs later as having -500 ppm. It seems the second number is rounded? Perhaps it is better to use a consistent rounded or unrounded value so as not to confuse the reader.

This has been corrected so that all shifts are now reported by their correct value of -452 ppm.

Red and magenta are difficult to tell apart for Bi vs Te in the TEM figure etc. Perhaps reconsider using colors that are more different.

The red colour in Figure 2 has been replaced with brown in order to improve readability.

The authors sometimes label axes with "arbitrary units" and sometimes omit this label, e.g. 2b vs 2c,d. Perhaps consider a blanket statement or label all axes appropriately...

The label (a.u.) has been omitted and described in the caption of Figure 2 for consistency.

Can the authors give a better explanation of the DFT vs experiment referencing procedure in the SM? My understanding is that a few NMR standards are used to extract a slope and intercept, and use this intercept as the "standard" that characterizes the difference between DFT and experiment. Can the authors elaborate on this a bit more (just one or two sentences).

The Reviewer is correct that this is a subtle point that requires more explanation. A full discussion has been added to the beginning of section 4 of the Supplementary Information regarding "Correlation between experimental isotropic ^{125}Te NMR chemical shift and the DFT-calculated isotropic magnetic shielding" (text highlighted yellow), which states the referencing procedure explicitly.

REVIEWERS' COMMENTS:

Reviewer #1 (Remarks to the Author):

I think the authors answered all of my comments and revised text accordingly. I recommend this revision for a publication.

Reviewer #2 (Remarks to the Author):

Referee Report - NCOMMS-19-34029 resub- Papawassiliou et al. - Resolving Dirac electrons with broadband high-resolution NMR: New perspectives in exploring topological quantum properties

The authors Papawassiliou et al. have carefully and reasonably responded to all of my questions and criticisms. The changes to the manuscript and updated interpretation of the data are reasonable and carefully worded such that the claims are not too strong. Considering these changes, I recommend publication in the current form with only minor copy editing corrections.

Just a few minor comments:

In the last sentence at the end of the section titled, "Resolving the bulk and Dirac edge states with aMAT NMR," on page 9, the authors cite Lee and Nagaosa and state that, "In Fermi liquids with similar topology,..." Are the authors referring to the topology of the orbital currents, band topology, or physical shape of the crystal. These are of course interrelated, but perhaps it could be worded more clearly.

Also, in the same sentence: in the clause "...which resembles with our observation." remove the word "with"

Reviewer 1 comments

Reviewer #1 (Remarks to the Author):

I think the authors answered all of my comments and revised text accordingly. I recommend this revision for a publication.

We are grateful to the reviewer for his/her comments on the previous draft, and for the final recommendation.

Reviewer 2 comments

Reviewer #2 (Remarks to the Author):

Referee Report - NCOMMS-19-34029 resub- Papawassiliou et al. - Resolving Dirac electrons with broadband high-resolution NMR: New perspectives in exploring topological quantum properties

The authors Papawassiliou et al. have carefully and reasonably responded to all of my questions and criticisms. The changes to the manuscript and updated interpretation of the data are reasonable and carefully worded such that the claims are not too strong. Considering these changes, I recommend publication in the current form with only minor copy editing corrections.

We thank the reviewer for his/her comments.

Just a few minor comments:

In the last sentence at the end of the section titled, "Resolving the bulk and Dirac edge states with aMAT NMR," on page 9, the authors cite Lee and Nagaosa and state that, "In Fermi liquids with similar topology,..." Are the authors referring to the topology of the orbital currents, band topology, or physical shape of the crystal. These are of course interrelated, but perhaps it could be worded more clearly.

The sentence has been reworded to clarify that we are referring to the topology of the orbital currents.

Also, in the same sentence: in the clause "...which resembles with our observation." remove the word "with"

This typo has been corrected.